# MUTUAL INFORMATION REGULARIZED OFFLINE REINFORCEMENT LEARNING

**Xiao Ma,**[*] **Bingyi Kang**[*◇]**, Zhongwen Xu, Min Lin, Shuicheng Yan**
Sea AI Lab
{max,kangby,xuzw,linmin,yansc}@sea.com

## ABSTRACT

Offline reinforcement learning (RL) aims at learning an effective policy from offline datasets without active interactions with the environment. The major challenge of offline RL is the distribution shift that appears when out-of-distribution actions are queried, which makes the policy improvement direction biased by extrapolation errors. Most existing methods address this problem by penalizing the policy for deviating from the behavior policy during policy improvement or making conservative updates for value functions during policy evaluation. In this work, we propose a novel MISA framework to approach offline RL from the perspective of **M**utual **I**nformation between **S**tates and **A**ctions in the dataset by directly constraining the policy improvement direction. Intuitively, mutual information measures the mutual dependence of actions and states, which reflects how a behavior agent reacts to certain environment states during data collection. To effectively utilize this information to facilitate policy learning, MISA constructs lower bounds of mutual information parameterized by the policy and Q-values. We show that optimizing this lower bound is equivalent to maximizing the likelihood of a one-step improved policy on the offline dataset. In this way, we constrain the policy improvement direction to lie in the data manifold. The resulting algorithm simultaneously augments the policy evaluation and improvement by adding a mutual information regularization. MISA is a general offline RL framework that unifies conservative Q-learning (CQL) and behavior regularization methods (*e.g.*, TD3+BC) as special cases. Our experiments show that MISA performs significantly better than existing methods and achieves new state-of-the-art on various tasks of the D4RL benchmark.

## 1 INTRODUCTION

Reinforcement learning (RL) has made remarkable achievements for solving sequential decision-making problems, ranging from game playing (Mnih et al., 2013; Silver et al., 2017; Berner et al., 2019) to robot control (Levine et al., 2016; Kahn et al., 2018; Savva et al., 2019). However, its success heavily relies on 1) an environment to interact with for data collection and 2) an online algorithm to improve the agent based only on its own trial-and-error experiences. These make RL algorithms incapable in real-world safety-sensitive scenarios where interactions with the environment are dangerous or prohibitively expensive, such as in autonomous driving and robot manipulation with human autonomy (Levine et al., 2020; Kumar et al., 2020). Therefore, offline RL is proposed to study the problem of learning decision-making agents from experiences that are previously collected from other agents when interacting with the environment is costly or not allowed.

Though much demanded, extending RL algorithms to offline datasets is challenged by the distributional shift between the data-collecting policy and the learning policy. Specifically, a typical RL algorithm alternates between evaluating the Q values of a policy and improving the policy to have better cumulative return under the current value estimation. When it comes to the offline setting, policy improvement often involves querying out-of-distribution (OOD) state-action pairs that have never appeared in the dataset, for which the Q values are over-estimated due to extrapolation error

---

[*]equal contribution, ◇ corresponding author

of neural networks. As a result, the policy improvement direction is erroneously affected, eventually leading to catastrophic explosion of value estimations as well as policy collapse after error accumulation. Existing methods (Kumar et al., 2020; Wang et al., 2020; Fujimoto & Gu, 2021; Yu et al., 2021) tackle this problem by either forcing the learned policy to stay close to the behavior policy (Fujimoto et al., 2019; Wu et al., 2019; Fujimoto & Gu, 2021) or generating low value estimations for OOD actions (Nachum et al., 2017; Kumar et al., 2020; Yu et al., 2021). Though these methods are effective at alleviating the distributional shift problem of the learning policy, the improved policy is unconstrained and might still deviate from the data distribution. A natural question thus arises: can we directly constrain the policy improvement direction to lie in the data manifold?

In this paper, we step back and consider the offline dataset from a new perspective, *i.e.*, the **M**utual **I**nformation between **S**tates and **A**ctions (MISA). By viewing state and action as two random variables, the mutual information represents the reduction of uncertainty of actions given certain states, *a.k.a.*, information gain in information theory (Nowozin, 2012). Therefore, mutual information is an appealing metric to sufficiently acquire knowledge from a dataset and characterize a behavior policy. We for the first time introduce it into offline RL as an regularization that directly constrains the policy improvement direction. Specifically, to allow practical optimizations of state-action mutual information estimation, we introduce the MISA lower bound of state-action pairs, which connects mutual information with RL by treating a parameterized policy as a variational distribution and the Q-values as the energy functions. We show that this lower bound can be interpreted as the likelihood of a non-parametric policy on the offline dataset, which actually represents the one-step improvement of the current policy based on the current value estimation. Maximizing MISA lower bound is equivalent to directly regularizing the policy improvement within the dataset manifold. However, the constructed lower bound involves integration over a self-normalized energy-based distribution, whose gradient estimation is intractable. To alleviate this dilemma, Markov Chain Monte Carlo (MCMC) estimation is adopted to produce an unbiased gradient estimation for MISA lower bound.

Theoretically, MISA is a general framework for offline RL that unifies several existing offline RL paradigms including behavior regularization and conservative learning. As examples, we show that TD3+BC (Fujimoto & Gu, 2021) and CQL (Kumar et al., 2020) are degenerated cases of MISA. In our experiments, we demonstrate that MISA achieves significantly better performance on various environments of the D4RL (Fu et al., 2020) benchmark than the state-of-the-art methods. Additional ablation studies, visualizations, and limitations are discussed to better understand the proposed method.

## 2   MUTUAL INFORMATION REGULARIZED OFFLINE RL

In this paper, we propose to think the offline RL problem from the perspective of mutual information and develop a novel framework (MISA) by estimating the **M**utual **I**nformation between **S**tates and **A**ctions of a given offline dataset. We show that MISA is a general framework which unifies multiple existing offline RL algorithms as special cases, including standard behavior cloning, TD3+BC (Fujimoto & Gu, 2021), and CQL (Kumar et al., 2020).

### 2.1   MUTUAL INFORMATION REGULARIZATION

Consider the state $S$ and action $A$ as two random variables. Let $p_{(S,A)}(s, a)$ denote the joint distribution of state-action pairs, and $p_S(s)$, $p_A(a)$ be the marginal distributions. The subscripts are omitted in the following for simplicity. The mutual information between $S$ and $A$ is defined with:

$$I(S; A) = \mathbb{E}_{p(s,a)} \left[ \log \frac{p(s, a)}{p(s)p(a)} \right] = \mathbb{E}_{p(s,a)} \left[ \log \frac{p(a \mid s)}{p(a)} \right] = H(A) - H(A \mid S), \quad (1)$$

where $H$ is Shannon entropy, and $H(A|S)$ is conditional entropy of $A$ given $S$. The higher mutual information between $S$ and $A$ means the lower uncertainty in $A$ given state $S$. This coincides with the observation that the actions selected by a well-performing agent are usually coupled with certain states. Therefore, given a joint distribution of state-action pairs induced from a (sub-optimal) behavior agent, it is natural to learn a policy that can recover the dependence between states and actions produced by the behavior agent. By regularizing the agent with $I(S; A)$ estimation, we encourage the agent to 1) perform policy update within the dataset distribution and 2) avoid being over-conservative and make sufficient use of the dataset information.

Let $\pi_\beta(a|s)$ represent a behavior policy and $p_\beta(s, a)$ be the joint distribution of state-action pairs induced by $\pi_\beta$. Calculating the mutual information is often intractable as accessing to $p_\beta(s, a)$ is infeasible. Fortunately, in the problem of offline reinforcement learning, a dataset $\mathcal{D} = \{(s_t, a_t, r_t, s_{t+1})\}$ of transitions is given by drawing samples independently from $p_\beta(s, a)$. This dataset can thus be seen as a sample-based empirical joint distribution $p_\mathcal{D}(s, a)$ for $p_\beta$. Let $\mathcal{I}(\theta, \phi)$ denote a mutual information lower bound that relies on parameterized functions with parameters $\theta$ and $\phi^1$, which are usually the policy network and Q network in the context of RL. We defer the derivation of such bounds in Sec. 2.2. Based on the above motivation, we aim at learning a policy that can approximate the mutual information of the dataset while being optimized to get the best possible cumulative return. We focus on the actor-critic framework, and formulate the offline RL problem with mutual information regularization as follows:

$$\min_{\phi} \quad \mathbb{E}_{s,a,s'\sim\mathcal{D}} \left[ \frac{1}{2} \left( Q_\phi(s, a) - \mathcal{B}^{\pi_\theta} Q_\phi(s, a) \right)^2 \right] - \alpha_1 \hat{\mathcal{I}}_\mathcal{D}(\theta, \phi), \quad \text{(Policy Evaluation)} \quad (2)$$

$$\max_{\theta} \quad \mathbb{E}_{s\sim\mathcal{D},a\sim\pi_\theta(a|s)} \left[ Q_\phi(s, a) \right] + \alpha_2 \hat{\mathcal{I}}_\mathcal{D}(\theta, \phi), \quad \text{(Policy Improvement)} \quad (3)$$

where $\alpha_1$ and $\alpha_2$ are the coefficients to balance RL objective and mutual information objective, and $\hat{\mathcal{I}}_\mathcal{D}(\theta, \phi)$ denotes the sample-based version of $\mathcal{I}(\theta, \phi)$ estimated from dataset $\mathcal{D}$.

## 2.2 STATE-ACTION MUTUAL INFORMATION ESTIMATION

In this section, we develop practical solutions to approximate the mutual information $I(S; A)$ from samples of the joint distribution. We use the learning policy $\pi_\theta(a|s)$ as a variational variable and Eqn. 1 can be rewritten as:

$$I(S; A) = \mathbb{E}_{p(s,a)} \left[ \log \frac{\pi_\theta(a|s)p(a|s)}{p(a)\pi_\theta(a|s)} \right] = \mathbb{E}_{p(s,a)} \left[ \log \frac{\pi_\theta(a|s)}{p(a)} \right] + D_{\text{KL}} \left( p(s, a) || p(s)\pi_\theta(a|s) \right), \quad (4)$$

where $p(s)\pi_\theta(a|s)$ is an induced joint distribution. Let $\mathcal{I}_{\text{BA}} \triangleq \mathbb{E}_{p(s,a)} \left[ \log \frac{\pi_\theta(a|s)}{p(a)} \right]$. We have $I(S; A) \geq \mathcal{I}_{\text{BA}}$ as the KL divergence is always non-negative. This is exactly the Barber-Agakov (BA) lower bound developed by (Barber & Agakov, 2004).

To obtain tighter bounds, we turn to KL dual representations of $D_{\text{KL}} \left( p(s, a) || p(s)\pi_\theta(a|s) \right)$ in Eqn. 4. To this end, we choose $\mathcal{F}$ to be a set of parameterized functions $T_\phi : S \times A \to \mathbb{R}, \phi \in \Phi$, which can be seen as an energy function.

With the $f$-divergence dual representation, we derive MISA-$f$ as

$$\mathcal{I}_{\text{MISA-}f} \triangleq \mathbb{E}_{p(s,a)} \left[ \log \frac{\pi_\theta(a|s)}{p(a)} \right] + \mathbb{E}_{p(s,a)} \left[ T_\phi(s, a) \right] - \mathbb{E}_{p(s)\pi_\theta(a|s)} \left[ e^{T_\phi(s,a)-1} \right]. \quad (5)$$

The $\mathcal{I}_{\text{MISA-}f}$ bound is tight when $p(a|s) \propto \pi_\theta(a|s)e^{T_\phi(s,a)-1}$. Similarly, using the DV representation in Theorem B.2, we can have another bound $\mathcal{I}_{\text{MISA-DV}} \leq I(S; A)$, as shown below:

$$\mathcal{I}_{\text{MISA-DV}} \triangleq \mathbb{E}_{p(s,a)} \left[ \log \frac{\pi_\theta(a|s)}{p(a)} \right] + \mathbb{E}_{p(s,a)} \left[ T_\phi(s, a) \right] - \log \mathbb{E}_{p(s)\pi_\theta(a|s)} \left[ e^{T_\phi(s,a)} \right], \quad (6)$$

which is tight when $p(a|s) = \frac{1}{\mathcal{Z}} p(s)\pi_\theta(a|s)e^{T_\phi(s,a)}$, where $\mathcal{Z} = \mathbb{E}_{p(s)\pi_\theta(a|s)} \left[ e^{T_\phi(s,a)} \right]$.

We observe that the KL term in Eqn. 4 can be rewritten as:

$$D_{\text{KL}} \left( p(s, a) || p(s)\pi_\theta(a|s) \right) = \mathbb{E}_{p(s)} \left[ \mathbb{E}_{p(a|s)} \left[ \log \frac{p(a|s)}{\pi_\theta(a|s)} \right] \right] = \mathbb{E}_{p(s)} \left[ D_{\text{KL}}(p(a|s)||\pi_\theta(a|s)) \right].$$

Applying the DV representation of $D_{\text{KL}}(p(a|s)||\pi_\theta(a|s))$, we can have a new lower bound $\mathcal{I}_{\text{MISA}}$:

$$\mathcal{I}_{\text{MISA}} \triangleq \mathbb{E}_{p(s,a)} \left[ \log \frac{\pi_\theta(a|s)}{p(a)} \right] + \mathbb{E}_{p(s,a)} \left[ T_\phi(s, a) \right] - \mathbb{E}_{p(s)} \log \mathbb{E}_{\pi_\theta(a|s)} \left[ e^{T_\phi(s,a)} \right]. \quad (7)$$

The bound is tight when $p(a|s) = \frac{1}{\mathcal{Z}(s)} \pi_\theta(a|s)e^{T_\phi(s,a)}$, where $\mathcal{Z}(s) = \mathbb{E}_{\pi_\theta(a|s)}[e^{T_\phi(s,a)}]$.

---

[1] Note some lower bounds might only have one parameterized function.

**Theorem 2.1.** *Given the joint distribution of state s and action a, the lower bounds of mutual information $I(S; A)$ defined in Eqn. 5-7 have the following relations:*

$$I(S; A) \geq \mathcal{I}_{MISA} \geq \mathcal{I}_{MISA\text{-}DV} \geq \mathcal{I}_{MISA\text{-}f}. \tag{8}$$

The proof is deferred to the appendix due to space limit.

## 2.3 Integration with Offline Reinforcement Learning

We now describe how our MISA lower bound is integrated into the above framework (Eqn. 9-10) to give a practical offline RL algorithm. We propose to use a Q network $Q_\phi(s, a)$ as the energy function $T_\phi(s, a)$, and use $p_\mathcal{D}(s, a)$ as the joint distribution in Eqn. 7. Then we have the following objective to learn a Q-network during policy evaluation:

$$J_Q(\phi) = J_Q^\mathcal{B}(\phi) - \gamma_1 \left( \mathbb{E}_{s,a\sim\mathcal{D}} \left[ Q_\phi(s, a) \right] - \mathbb{E}_{s\sim\mathcal{D}} \left[ \log \mathbb{E}_{\pi_\theta(a|s)} \left[ e^{Q_\phi(s,a)} \right] \right] \right), \tag{9}$$

where $J_Q^\mathcal{B}(\phi) = \mathbb{E}_{s,a,s'\sim\mathcal{D}} \left[ \frac{1}{2} \left( Q_\phi(s, a) - \mathcal{B}^{\pi_\theta} Q_\phi(s, a) \right)^2 \right]$ represents the TD error. For policy improvement, note that the entropy term $H(a)$ in Eqn. 7 can be omitted as it is a constant given dataset $\mathcal{D}$. Thus, we have the below objective to maximize:

$$J_\pi(\theta) = \mathbb{E}_{s\sim\mathcal{D},a\sim\pi_\theta(a|s)} \left[ Q_\phi(s, a) \right] + \gamma_2 \left( \mathbb{E}_{s,a\sim\mathcal{D}}[\log \pi_\theta(a|s)] - \mathbb{E}_{s\sim\mathcal{D}} \left[ \log \mathbb{E}_{\pi_\theta(a|s)} \left[ e^{Q_\phi(s,a)} \right] \right] \right). \tag{10}$$

The formulations for other regularizers (*e.g.*, $\mathcal{I}_{\text{MISA-DV}}$ and $\mathcal{I}_{\text{MISA-}f}$) can be derived similarly. A detailed description of the MISA algorithm for offline RL can be found in Algo. 1.

**Intuitive Explanation on the Mutual Information Regularizer.** By rearranging the terms in Eqn. 7, MISA can be written as:

$$\mathcal{I}_{\text{MISA}} = \mathbb{E}_{s,a\sim\mathcal{D}} \left[ \log \frac{\pi_\theta(a \mid s)e^{Q_\phi(s,a)}}{\mathbb{E}_{\pi_\theta(a'|s)} \left[ e^{Q_\phi(s,a')} \right]} \right], \tag{11}$$

where the log term can be seen as the log probability of a one-step improved policy. More specifically, for policy improvement with KL divergence regularization: $\max_\pi \mathbb{E}_{s\sim\mathcal{D},a\sim\pi}[Q_\phi(s, a)] + D_{\text{KL}}(\pi||\pi_\theta)$, the optimal solution is given by $\pi_{\theta,\phi}^* \propto \pi_\theta(a|s)e^{Q_\phi(s,a)}$ (Abdolmaleki et al., 2018; Peng et al., 2019). Therefore, $\mathcal{I}_{\text{MISA}}$ is rewritten with $\pi_{\theta,\phi}^*$ as $\mathcal{I}_{\text{MISA}} = \mathbb{E}_{s,a\sim\mathcal{D}}[\log \pi_{\theta,\phi}^*(a|s)]$, and maximizing it means maximizing the log likelihood of the dataset using the improved policy. In other words, instead of directly fitting the policy on the dataset, which is short-sighted, this objective considers the optimization direction of the policy improvement step. Given the current policy and policy evaluation results, it first computes the analytic improved policy, and then forces the dataset likelihood to be maximized using the improved policy. In this way, even if an out-of-distribution state-action pair get an overestimated q value, $\mathcal{I}_{\text{MISA}}$ is going to suppress this value and make sure in-distribution data have relatively higher value estimation.

**Unbiased Gradient Estimation** For policy improvement with Eqn. 10, differentiating through a sampling distribution $\pi_\theta(a \mid s)$ is required for $\mathbb{E}_{s\sim D} \log \mathbb{E}_{\pi_\theta(a|s)} \left[ e^{Q_\phi(s,a)} \right]$. For a Gaussian policy $\pi_\theta(a \mid s) = \mathcal{N}(\mu_\theta, \sigma_\theta)$, one could consider the reparameterization trick (Kingma & Welling, 2014) and convert the objective as $\mathbb{E}_{s\sim D} \log \mathbb{E}_{\epsilon\sim\mathcal{N}(0,\mathbb{I})} \left[ e^{Q_\phi(s,\mu_\theta+\epsilon*\sigma_\theta)} \right]$. However, this introduces high variance in offline reinforcement learning setups because we condition the policy improvement directly on the Q values of the out-of-distribution actions, which eventually gives a noisy policy. Hence, we aim to minimize the influence of Q values for policy improvement.

Differentiating Eqn. 7 with respect to policy parameters $\theta$, we have

$$\frac{\partial \mathcal{I}_{\text{MISA}}}{\partial \theta} = \mathbb{E}_{s,a\sim D} \left[ \frac{\log \pi_\theta(a \mid s)}{\partial \theta} \right] - \mathbb{E}_{s\sim D,a\sim p_{\theta,\phi}(a|s)} \left[ \frac{\log \pi_\theta(a \mid s)}{\partial \theta} \right] \tag{12}$$

where $p_{\theta,\phi}(a \mid s) = \frac{\pi_\theta(a|s)e^{Q_\phi(s,a)}}{\mathbb{E}_{\pi_\theta(a|s)}\left[e^{Q_\phi(s,a)}\right]}$ is a self-normalized distribution. See appendix C.2 for a derivation. By optimizing Eqn. 12, we obtain an unbiased gradient estimation of the MISA objective with respect to the policy parameters, while minimizing the negative effects of the Q values of OOD actions. To sample from $p_{\theta,\phi}(a \mid s)$, one can consider Markov-Chain Monte-Carlo (MCMC) methods, e.g., Hamiltonian Monte Carlo (Betancourt, 2017).

| Dataset | BC | 10%BC | DT | AWAC | OneStep RL | TD3+BC | CQL | IQL | MISA |
|---|---|---|---|---|---|---|---|---|---|
| halfcheetah-medium-v2 | 42.6 | 42.5 | 42.6 | 43.5 | **48.4** | 48.3 | 44 | 47.4 | 47.4 |
| hopper-medium-v2 | 52.9 | 56.9 | **67.6** | 57 | 59.6 | 59.3 | 58.5 | 66.3 | 67.1 |
| walker2d-medium-v2 | 75.3 | 75 | 74 | 72.4 | 81.8 | 83.7 | 72.5 | 78.3 | **84.1** |
| halfcheetah-medium-replay-v2 | 36.6 | 40.6 | 36.6 | 40.5 | 38.1 | 44.6 | 45.5 | 44.2 | **45.6** |
| hopper-medium-replay-v2 | 18.1 | 75.9 | 82.7 | 37.2 | 97.5 | 60.9 | 95 | 94.7 | **98.6** |
| walker2d-medium-replay-v2 | 26 | 62.5 | 66.6 | 27 | 49.5 | 81.8 | 77.2 | 73.9 | **86.2** |
| halfcheetah-medium-expert-v2 | 55.2 | 92.9 | 86.8 | 42.8 | 93.4 | 90.7 | 91.6 | 86.7 | **94.7** |
| hopper-medium-expert-v2 | 52.5 | **110.9** | 107.6 | 55.8 | 103.3 | 98 | 105.4 | 91.5 | 109.8 |
| walker2d-medium-expert-v2 | 107.5 | 109 | 108.1 | 74.5 | **113** | 110.1 | 108.8 | 109.6 | 109.4 |
| gym-locomotion-v2 (total) | 466.7 | 666.2 | 672.6 | 450.7 | 684.6 | 677.4 | 698.5 | 692.6 | **742.9** |
| kitchen-complete-v0 | 65 | - | - | - | - | - | 43.8 | 62.5 | **70.2** |
| kitchen-partial-v0 | 38 | - | - | - | - | - | **49.8** | 46.3 | 45.7 |
| kitchen-mixed-v0 | 51.5 | - | - | - | - | - | 51 | 51 | **56.6** |
| kitchen-v0 (total) | 154.5 | - | - | - | - | - | 144.6 | 159.8 | **172.5** |
| pen-human-v0 | 63.9 | - | - | - | - | - | 37.5 | 71.5 | **88.1** |
| hammer-human-v0 | 1.2 | - | - | - | - | - | 4.4 | 1.4 | **8.1** |
| door-human-v0 | 2 | - | - | - | - | - | **9.9** | 4.3 | 5.2 |
| relocate-human-v0 | 0.1 | - | - | - | - | - | 0.2 | 0.1 | 0.1 |
| pen-cloned-v0 | 37 | - | - | - | **60** | - | 39.2 | 37.3 | 58.6 |
| hammer-cloned-v0 | 0.6 | - | - | - | 2.1 | - | 2.1 | 2.1 | **2.2** |
| door-cloned-v0 | 0 | - | - | - | 0.4 | - | 0.4 | **1.6** | 0.5 |
| relocate-cloned-v0 | -0.3 | - | - | - | -0.1 | - | -0.1 | -0.2 | -0.1 |
| adroit-v0 (human+cloned) | 104.5 | 0 | 0 | 0 | 62.4 | 0 | 93.6 | 118.1 | **162.7** |
| antmaze-umaze-v0 | 54.6 | 62.8 | 59.2 | 56.7 | 64.3 | 78.6 | 74 | 87.5 | **92.3** |
| antmaze-umaze-diverse-v0 | 45.6 | 50.2 | 53 | 49.3 | 60.7 | 71.4 | 84 | 62.2 | **89.1** |
| antmaze-medium-play-v0 | 0 | 5.4 | 0 | 0 | 0.3 | 10.6 | 61.2 | **71.2** | 63 |
| antmaze-medium-diverse-v0 | 0 | 9.8 | 0 | 0.7 | 0 | 3 | 53.7 | **70** | 62.8 |
| antmaze-large-play-v0 | 0 | 0 | 0 | 0 | **0** | 0.2 | 15.8 | **39.6** | 17.5 |
| antmaze-large-diverse-v0 | 0 | 6 | 0 | 1 | 0 | 0 | 14.9 | **47.5** | 23.4 |
| antmaze-v0 (total) | 100.2 | 134.2 | 112.2 | 107.7 | 125.3 | 163.8 | 303.6 | **378** | 348.1 |

Table 1: Average normalized score on the D4RL benchmark. Results of baselines are taken directly from (Kostrikov et al., 2022).

## 3 SUMMARY OF EXPERIMENTS

We conduct extensive experiments on various tasks of the D4RL benchmark (Fu et al., 2020). We show that MISA generally outperforms the state-of-the-art methods on these domains. The results are presented in Table 1. Qualitatively, MISA learns a good latent representation of offline dataset that disentangles the state-action pairs with high rewards. In addition, we ablate MISA on its parameters and variants to better understand the contributing components. Our results suggest that a tighter mutual information lower bound generally improves the performance on offline RL. For more details, we refer to our appendix.

## 4 CONCLUSIONS

We present the MISA framework for offline reinforcement learning by directly regularizing policy improvement and policy evaluation with the mutual information between state-action pairs of the dataset. MISA connects mutual information estimation with RL by constructing tractable lower bounds, treating the learning policy as a variational distribution and Q values as energy functions. The resulting tractable lower bound resembles a non-parametric energy-based distribution, which can be interpreted as the likelihood of a one-step improved policy given current value estimation. In this way, MISA can constrain the policy improvement within the dataset manifold. In our experiments, MISA significantly outperforms the state-of-the-art methods on D4RL benchmark. However, MISA assumes a high correspondence between states and actions, which might fail on uncorrelated data generated by a random policy. We leave it for future study.

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

## A    RELATED WORKS

**Offline Reinforcement Learning**    The most critical challenge for extending an off-policy RL algorithm to an offline setup is the distribution shift between the behavior policy, *i.e.*, the policy for data collection, and the learning policy. To tackle this challenge, most of the offline RL algorithms consider a conservative learning framework. They either regularize the learning policy to stay close to the behavior policy (Fujimoto et al., 2019; Wu et al., 2019; Fujimoto & Gu, 2021; Siegel et al., 2020; Wang et al., 2020), or force Q values to be low for OOD state-action pairs (Nachum et al., 2017; Kumar et al., 2020; Yu et al., 2021). For example, TD3+BC (Fujimoto & Gu, 2021) adds an additional behavior cloning (BC) signal along with the TD3 (Fujimoto et al., 2018), which encourages the policy to stay in the data manifold; CQL (Kumar et al., 2020), from the Q-value perspective, penalizes the OOD state-action pairs for generating high Q-value estimations and learns a lower bound of the true value function. However, their policy improvement direction is unconstrained and might deviate from the data distribution. On the other hand, SARSA-style updates (Sutton & Barto, 2018) are considered to only query in-distribution state-action pairs (Peng et al., 2019; Kostrikov et al., 2022). Nevertheless, without explicitly querying Bellman's optimality equation, they limit the policy from producing unseen actions. Our proposed MISA follows the conservative framework and directly regularizes the policy improvement direction to lie within the data manifold with mutual information, which more fully exploits the dataset information while learning a conservative policy.

**Mutual Information Estimation.**    Mutual information is a fundamental quantity in information theory, statistics, and machine learning. However, direct computation of mutual information is intractable as it involves computing a log partition function of a high dimensional variable. Thus, how to estimate the mutual information $I(x, z)$ between random variables $\mathcal{X}$ and $\mathcal{Z}$, accurately and efficiently, is a critical issue. One straightforward lower bound for mutual information estimation is Barber-Agakov bound (Barber & Agakov, 2004), which introduces an additional variational distribution $q(z \mid x)$ to approximate the unknown posterior $p(z \mid x)$. Instead of using an explicit "decoder" $q(z \mid x)$, we can use *unnormalized* distributions for the variational family $q(z \mid x)$ (Donsker & Varadhan, 1975; Belghazi et al., 2018; Oord et al., 2018), i.e., approximate the distribution as $q(z \mid x) = \frac{p(z)e^{f(x,z)}}{\mathbb{E}_{p(z)}[e^{f(x,z)}]}$, where $f(x, z)$ is an arbitrary critic function. As an example, InfoNCE (Oord et al., 2018) has been widely used in representation learning literature (Oord et al., 2018; He et al., 2020; Chen et al., 2020). To further improve the mutual information estimation, a combination of normalized and unnormalized variational distribution family can be considered (Brekelmans et al., 2022; Poole et al., 2019). Our MISA connects mutual information estimation with RL by parameterizing a tractable lower bound with a policy network as a variational distribution and the Q values as critics. In this way, MISA explicitly regularizes the policy improvement direction to lie in the data manifold and produces strong empirical performance.

## B    PRELIMINARIES

**Reinforcement Learning**    We consider a Markov Decision Process (MDP) denoted as a tuple $\mathcal{M} = (\mathcal{S}, \mathcal{A}, p_0(s), p(s' \mid s, a), r(s, a), \gamma)$, where $\mathcal{S}$ is the state space, $\mathcal{A}$ is the action space, $p_0(s)$ is the initial state distribution, $p(s' \mid s, a)$ is the transition function, $r(s, a)$ is the reward function, and $\gamma$ is the discount factor. The target of a learning agent is to find a policy $\pi^*(a \mid s)$ that maximizes the accumulative reward by interacting with the environment

$$\pi^* = \arg \max_{\pi} \mathbb{E}_{\pi} \left[ \sum_{t=0}^{\infty} \gamma^t r(s_t, a_t) \mid s_0 \sim p_0(s), a_t \sim \pi(a \mid s_t) \right]. \tag{13}$$

Q-learning is a set of off-policy RL algorithms that utilize the optimal Bellman's optimality operator $\mathcal{B}^* Q(s, a) = r(s, a) + \gamma \mathbb{E}_{s' \sim p(s'|s,a)}[\max_{a'} Q(s', a')]$ to learn a Q function. Differently, Bellman's expectation operator $\mathcal{B}^\pi Q(s, a) = r(s, a) + \gamma \mathbb{E}_{s' \sim p(s'|s,a); a' \sim \pi(\cdot|s')}[Q(s', a')]$ gives an actor-critic framework that alternates between policy evaluation and policy improvement. Consider a value network $Q_\phi(s, a)$ parameterized by $\phi$ and a policy network $\pi_\theta(a|s)$ parameterzied by $\theta$. Let $\mu_\pi(s)$ denote the stationary distribution induced with policy $\pi$, which is also called occupancy measure (Schulman et al., 2015). Given the current policy, the policy evaluation aims to learn a Q network that can accurately predict its values minimizing $\mathbb{E}_{\mu_{\pi_\theta}(s)\pi_\theta(a|s)}[(Q_\phi(s, a) - \mathcal{B}^{\pi_\theta} Q_\phi(s, a))^2]$.

Policy improvement focuses on learning the optimal policy that maximizes $\mathbb{E}_{\mu_\pi(s)\pi(a|s)}[Q_\phi(s,a)]$. In practical implementations, the Bellman operator is usually replaced with its sample-based version $\hat{\mathcal{B}}$, and the expectation over $\mu_\pi(s)\pi(a|s)$ is approximated by an online replay buffer or an offline dataset $\mathcal{D}$.

Nevertheless, as it is unavoidable to query the OOD actions when performing the maximization over actions, an inaccurate over-estimation of Q value will be selected and the error will accumulate during the Bellman's update. Conservative RL methods, in turn, aim to perform "conservative" updates of the value / policy function during optimization by constraining the updates on only the in-distribution samples, which eventually minimizes the negative impact of OOD actions.

**KL Divergence** Given two probability distributions $p(x)$ and $q(x)$ on the same probability space, the KL divergence (*i.e.*, relative entropy) from $q$ to $p$ is given by $D_{\text{KL}}(p||q) = \mathbb{E}_{p(x)}\left[\log \frac{p(x)}{q(x)}\right] \geq 0$. The minimum value is achieved when the two densities are identical. We consider two dual representations that result in tractable estimators for the KL divergence.

**Lemma B.1** (*f*-divergence representation (Nowozin et al., 2016))**.** *The KL divergence admits the following lower bound:*

$$D_{KL}(p||q) \geq \sup_{T \in \mathcal{F}} \mathbb{E}_{p(x)}[T(x)] - \mathbb{E}_{q(x)}[e^{T(x)-1}], \tag{14}$$

*where the supremum is taken over a function family $\mathcal{F}$ satisifying the intergrability constraints.*

**Lemma B.2** (Donsker-Varadhan representation (Nguyen et al., 2010))**.** *The KL divergence has the lower bound:*

$$D_{KL}(p||q) \geq \sup_{T \in \mathcal{F}} \mathbb{E}_{p(x)}[T(x)] - \log(\mathbb{E}_{q(x)}[e^{T(x)}]), \tag{15}$$

*where the supremum is taken over a function family $\mathcal{F}$ satisifying the intergrability constraints.*

The above two bounds are tight for sufficiently large families $\mathcal{F}$.

---

**Algorithm 1** Mutual Information Regularized Offline RL

---

**Input**: Initialize Q network $Q_\phi$, policy network $\pi_\theta$, dataset $\mathcal{D}$, hyperparameters $\alpha_1$ and $\alpha_2$.
**for** $t \in \{1, \ldots, \text{MAX\_STEP}\}$ **do**
    Train the Q network by gradient descent with objective $J_Q(\phi)$ in Eqn. 9:
    $\phi := \phi - \eta_Q \nabla_\phi J_Q(\phi)$
    Improve policy network by gradient ascent with object $J_\pi(\theta)$ in Eqn. 10:
    $\theta := \theta + \eta_\pi \nabla_\theta \mathbb{E}_{s \sim \mathcal{D}, a \sim \pi_\theta(a|s)}[Q_\phi(s,a)] + \alpha_2 \nabla_\theta I_{\text{MISA}}$
**end**
**Output**: The well-trained $\pi_\theta$.

---

## C  PROOFS AND DERIVATIONS

### C.1  PROOF FOR THEOREM 2.1

We first show $\mathcal{I}_{\text{MISA}}$, $\mathcal{I}_{\text{MISA-DV}}$ and $\mathcal{I}_{\text{MISA-}f}$ are lower bounds for mutual information $I(S,A)$.

Let $\mu_{\theta,\phi}(a|s) \triangleq \frac{1}{\mathcal{Z}(s)}\pi_\theta(a|s)e^{T_\phi(s,a)}$, where $\mathcal{Z}(s) = \mathbb{E}_{\pi_\theta(a|s)}[e^{T_\phi(s,a)}]$, $\mathcal{I}_{\text{MISA}}$ can be written as:

$$
\begin{aligned}
\mathcal{I}_{\text{MISA}} &\triangleq \mathbb{E}_{p(s,a)}\left[\log\frac{\pi_\theta(a|s)}{p(a)}\right] + \mathbb{E}_{p(s,a)}[T_\phi(s,a)] - \mathbb{E}_{p(s)}\log\mathbb{E}_{\pi_\theta(a|s)}\left[e^{T_\phi(s,a)}\right] \\
&= \mathbb{E}_{p(s,a)}\left[\log\frac{p(a|s)}{p(a)}\right] - \mathbb{E}_{p(s,a)}[\log p(a|s)] \\
&\quad + \mathbb{E}_{p(s,a)}[\log\pi_\theta(a|s)] + \mathbb{E}_{p(s,a)}[T_\phi(s,a)] - \mathbb{E}_{p(s)}[\log\mathcal{Z}(s)] \\
&= I(S,A) - \mathbb{E}_{p(s)}[D_{\text{KL}}(p(a|s)||\mu_{\theta,\phi}(a|s))] \leq I(S,A).
\end{aligned}
\tag{16}
$$

The above inequality holds as the KL divergence is always non-negative.

Similarly, let $\mu_{\theta,\phi}(s,a) \triangleq \frac{1}{\mathcal{Z}} p(s)\pi_\theta(a|s)e^{T_\phi(s,a)}$, where $\mathcal{Z}(s) = \mathbb{E}_{p(s)\pi_\theta(a|s)}[e^{T_\phi(s,a)}]$, $\mathcal{I}_{\text{MISA-DV}}$ can be written as:

$$
\begin{aligned}
\mathcal{I}_{\text{MISA-DV}} &\triangleq \mathbb{E}_{p(s,a)}\left[\log\frac{\pi_\theta(a|s)}{p(a)}\right] + \mathbb{E}_{p(s,a)}\left[T_\phi(s,a)\right] - \log\mathbb{E}_{p(s)\pi_\theta(a|s)}\left[e^{T_\phi(s,a)}\right] \\
&= \mathbb{E}_{p(s,a)}\left[\log\frac{p(a|s)}{p(a)}\right] - \mathbb{E}_{p(s,a)}[\log p(a|s)] \\
&\quad + \mathbb{E}_{p(s,a)}[\log\pi_\theta(a|s)] + \mathbb{E}_{p(s,a)}\left[T_\phi(s,a)\right] - \log\mathcal{Z} \\
&= I(S,A) - D_{\text{KL}}(p(s,a)||\mu_{\theta,\phi}(s,a)) \leq I(S,A).
\end{aligned}
\tag{17}
$$

The above inequality holds as the KL divergence is always non-negative.

Consider the generalized KL-divergence (Cichocki & Amari, 2010; Brekelmans et al., 2022) between two un-normalized distributions $\tilde{p}(x)$ and $\tilde{q}(x)$ defined by

$$
D_{\text{GKL}}(\tilde{p}(x)||\tilde{q}(x)) = \int \tilde{p}(x)\log\frac{\tilde{p}(x)}{\tilde{q}(x)} - \tilde{p}(x) + \tilde{q}(x)dx,
\tag{18}
$$

which is always non-negative and reduces to KL divergence when $\tilde{p}$ and $\tilde{q}$ are normalized. Let $\tilde{\mu}_{\theta,\phi}(a|s) \triangleq \pi_\theta(a|s)e^{T_\phi(s,a)-1}$ denote an un-normalized policy. We can rewrite $\mathcal{I}_{\text{MISA-}f}$ as

$$
\begin{aligned}
\mathcal{I}_{\text{MISA-}f} &\triangleq \mathbb{E}_{p(s,a)}\left[\log\frac{\pi_\theta(a|s)}{p(a)}\right] + \mathbb{E}_{p(s,a)}\left[T_\phi(s,a)\right] - \mathbb{E}_{p(s)\pi_\theta(a|s)}\left[e^{T_\phi(s,a)-1}\right] \\
&= \mathbb{E}_{p(s,a)}\left[\log\frac{p(a|s)}{p(a)}\right] - \mathbb{E}_{p(s,a)}[\log p(a|s)] \\
&\quad + \mathbb{E}_{p(s,a)}[\log\pi_\theta(a|s)] + \mathbb{E}_{p(s,a)}\left[T_\phi(s,a)-1\right] + 1 - \mathbb{E}_{p(s)\pi_\theta(a|s)}\left[e^{T_\phi(s,a)-1}\right] \\
&= I(S,A) - \mathbb{E}_{p(s)}\left[D_{\text{GKL}}(p(a|s)||\tilde{\mu}_{\theta,\phi}(a|s))\right] \leq I(S,A).
\end{aligned}
\tag{19}
$$

So far, we have proven that $\mathcal{I}_{\text{MISA}}$, $\mathcal{I}_{\text{MISA-DV}}$ and $\mathcal{I}_{\text{MISA-}f}$ mutual information lower bounds. Then we are going to prove their relations by starting fromt he relation between $\mathcal{I}_{\text{MISA}}$ and $\mathcal{I}_{\text{MISA-DV}}$.

$$
\begin{aligned}
\mathcal{I}_{\text{MISA}} - \mathcal{I}_{\text{MISA-DV}} &= D_{\text{KL}}(p(s,a)||\mu_{\theta,\phi}(s,a)) - \mathbb{E}_{p(s)}\left[D_{\text{KL}}(p(a|s)||\mu_{\theta,\phi}(a|s))\right] \\
&= \mathbb{E}_{p(s)}\mathbb{E}_{p(a|s)}\left[\log\frac{p(s,a)}{p(a|s)} - \log\frac{\mu_{\theta,\phi}(s,a)}{\mu_{\theta,\phi}(a|s)}\right] \\
&= \mathbb{E}_{p(s)}\mathbb{E}_{p(a|s)}\left[\log p(s) - \log\frac{1}{\mathcal{Z}}p(s)\mathcal{Z}(s)\right] \\
&= \mathbb{E}_{p(s)}\left[\log p(s) - \log\frac{1}{\mathcal{Z}}p(s)\mathcal{Z}(s)\right] \\
&= D_{\text{KL}}\left(p(s)||\frac{1}{\mathcal{Z}}p(s)\mathcal{Z}(s)\right) \geq 0,
\end{aligned}
\tag{20}
$$

where $\frac{1}{\mathcal{Z}}p(s)\mathcal{Z}(s)$ is a self-normalized distribution as $\mathcal{Z} = \mathbb{E}_{p(s)[\mathcal{Z}(s)]}$. Therefore, we have $\mathcal{I}_{\text{MISA}} \geq \mathcal{I}_{\text{MISA-DV}}$.

Similarly, the relation between $\mathcal{I}_{\text{MISA-DV}}$ and $\mathcal{I}_{\text{MISA-}f}$ is given by:

$$\mathcal{I}_{\text{MISA-DV}} - \mathcal{I}_{\text{MISA-}f} = \mathbb{E}_{p(s)}\left[D_{\text{GKL}}(p(a|s)||\tilde{\mu}_{\theta,\phi}(a|s))\right] - D_{\text{KL}}(p(s,a)||\mu_{\theta,\phi}(s,a))$$

$$= \mathbb{E}_{p(s)}\mathbb{E}_{p(a|s)}\left[\log\frac{p(a|s)}{p(s,a)} - \log\frac{\tilde{\mu}_{\theta,\phi}(a|s)}{\mu_{\theta,\phi}(s,a)}\right] - 1 + \mathbb{E}_{p(s)}\mathbb{E}_{\pi_\theta(a|s)}\left[e^{T_\phi(s,a)-1}\right]$$

$$= \mathbb{E}_{p(s)}\mathbb{E}_{p(a|s)}\left[-\log p(s) - \log\frac{\tilde{\mu}_{\theta,\phi}(a|s)}{\mu_{\theta,\phi}(s,a)}\right] - 1 + \mathbb{E}_{p(s)}\mathbb{E}_{\pi_\theta(a|s)}\left[e^{T_\phi(s,a)-1}\right]$$

$$= \mathbb{E}_{p(s)}\mathbb{E}_{p(a|s)}\left[\log\frac{\mu_{\theta,\phi}(s,a)}{p(s)\tilde{\mu}_{\theta,\phi}(a|s)}\right] - \mathbb{E}_{\mu_{\theta,\phi}(s,a)}[1] + \mathbb{E}_{p(s)}\mathbb{E}_{\pi_\theta(a|s)}\left[e^{T_\phi(s,a)-1}\right]$$

$$= \mathbb{E}_{p(s,a)}\left[\log\frac{e}{\mathcal{Z}}\right] - \mathbb{E}_{\mu_{\theta,\phi}(s,a)}[1] + \mathbb{E}_{p(s)}\mathbb{E}_{\pi_\theta(a|s)}\left[e^{T_\phi(s,a)-1}\right]$$

$$= \mathbb{E}_{\mu_{\theta,\phi}(s,a)}\left[\log\frac{e}{\mathcal{Z}}\right] - \mathbb{E}_{\mu_{\theta,\phi}(s,a)}[1] + \mathbb{E}_{p(s)}\mathbb{E}_{\pi_\theta(a|s)}\left[e^{T_\phi(s,a)-1}\right]$$

$$= \mathbb{E}_{\mu_{\theta,\phi}(s,a)}\left[\log\frac{\mu_{\theta,\phi}(s,a)}{p(s)\tilde{\mu}_{\theta,\phi}(a|s)}\right] - \mathbb{E}_{\mu_{\theta,\phi}(s,a)}[1] + \mathbb{E}_{p(s)}\mathbb{E}_{\pi_\theta(a|s)}\left[e^{T_\phi(s,a)-1}\right]$$

$$= D_{\text{GKL}}\left(\mu_{\theta,\phi}(s,a)||p(s)\tilde{\mu}_{\theta,\phi}(a|s)\right) \geq 0,$$

(21)

where $p(s)\tilde{\mu}_{\theta,\phi}(a|s)$ is an unnormalized joint distribution. Therefore, we have $I(S,A) \geq \mathcal{I}_{\text{MISA}} \geq \mathcal{I}_{\text{MISA-DV}} \geq \mathcal{I}_{\text{MISA-}f}$.

## C.2 DERIVATION OF MISA GRADIENTS

We detail how the unbiased gradient is derived in Sec.2.3.

$$\frac{\partial\mathcal{I}_{\text{MISA}}}{\partial\theta} = \mathbb{E}_{s,a\sim D}\left[\frac{\log\pi_\theta(a\mid s)}{\partial\theta}\right] - \mathbb{E}_{s\sim D}\left[\frac{\partial\log\mathbb{E}_{\pi_\theta(a|s)}[e^{Q_\phi(s,a)}]}{\partial\theta}\right]$$

$$= \mathbb{E}_{s,a\sim D}\left[\frac{\log\pi_\theta(a\mid s)}{\partial\theta}\right] - \mathbb{E}_{s\sim\mathcal{D}}\left[\mathbb{E}_{\pi_\theta(a|s)}\left[\frac{e^{Q_\phi(s,a)}}{\mathbb{E}_{\pi_\theta(a|s)}\left[e^{Q_\phi(s,a)}\right]}\frac{\log\pi_\theta(a\mid s)}{\partial\theta}\right]\right] \quad (22)$$

$$= \mathbb{E}_{s,a\sim D}\left[\frac{\log\pi_\theta(a\mid s)}{\partial\theta}\right] - \mathbb{E}_{s\sim D,a\sim p_{\theta,\phi}(a|s)}\left[\frac{\log\pi_\theta(a\mid s)}{\partial\theta}\right] \quad (23)$$

for Eqn. 22, we use the log-derivative trick.

## D CONNECTIONS TO EXISTING OFFLINE RL METHODS

We show that some existing offline RL methods can be viewed as special cases of MISA framework.

**Behavior Cloning and BC Regularized RL**  We first show that behavior cloning is a form of mutual information regularizer. As shown by Eqn. 4, $\mathcal{I}_{\text{BA}} \triangleq \mathbb{E}_{s,a\sim\mathcal{D}}\left[\log\frac{\pi_\theta(a|s)}{p(a)}\right]$ gives a lower bound of mutual information. Since $H(a)$ is a consistent given datasets, maximizing $\mathcal{I}_{\text{BA}}$ is equivalent to maximizing $\mathbb{E}_{s,a\sim\mathcal{D}}\left[\log\pi_\theta(a|s)\right]$, which is exactly the objective for behavior cloning.

As for TD3+BC (Fujimoto & Gu, 2021), the policy evaluation is unchanged, while the policy improvement objective is augmented by an MSE regularization term, i.e., $\mathbb{E}_{s\sim\mathcal{D}}[Q(s,\pi_\theta(s))] - \gamma\mathbb{E}_{s,a\sim\mathcal{D}}\left[(\pi_\theta(s)-a)^2\right]$, where $\lambda$ is a hyperparameter. Maximizing the negative MSE term is equivalent to maximizing $\mathbb{E}_{s,a\sim\mathcal{D}}\left[\log p_{\pi_\theta}(a|s)\right]$, where $p_{\pi_\theta} = Ce^{-\frac{1}{2}(\pi_\theta(s)-a)^2}$ is a Gaussian distribution, and $C$ is a constant. This is a special case of Eqn. 10 when we remove the last log-mean-exp term.

**Conservation Q Learning**  CQL (Kumar et al., 2020) was proposed to alleviate the overestimation issue of Q learning by making conservative updates to the Q values during policy evaluation. The policy improvement is kept unchanged compared to standard Q learning. We focus on the entropy-regularized policy evaluation of CQL as below:

$$\min_\phi J_Q^\mathcal{B}(\phi) - \gamma_1\mathbb{E}_{s\sim\mathcal{D}}\left[\mathbb{E}_{a\sim\pi_\mathcal{D}(a|s)}[Q_\phi(s,a)] - \log\sum_a e^{Q_\phi(s,a)}\right], \quad (24)$$

where we highlight the main difference between it and our MISA policy evaluation (Eqn. 9) in blue. Let $\pi_{\mathrm{U}}(a|s)$ denote a uniform distribution of actions and $|A|$ is the number of actions. The log-sum-exp term can be written as $\log \mathbb{E}_{a \sim \pi_{\mathrm{U}}(a|s)}[Q_\phi(s, a)] + \log |A|$. Substituting it into Eqn. 24 and discarding the constant $\log |A|$, we recover the the formulation in Eqn. 9. Therefore, CQL is actually doing mutual information regularization during policy evaluation. The key difference is that it is not using the current policy network as the variational distribution. Instead, a manually designed distribution is used in CQL. However, a uniform policy is usually suboptimal in environments with continuous actions. CQL thus constructs a mixed variational policy by drawing samples drawn from the current policy network, a uniform distribution and the dataset. In our formulation, the variational distribution will be optimized to give a better mutual information estimation. This might explain why MISA is able to give better performance than CQL.

## E  EXPERIMENTS

We perform extensive experiments on various tasks of the D4RL benchmark (Fu et al., 2020) to demonstrate effectiveness of the proposed method MISA. To provide better understandings of MISA, we provide additional ablation studies, visualizations, and discussions on the limitations.

### E.1  OFFLINE REINFORCEMENT LEARNING ON D4RL BENCHMARKS

**Experiment Setups.** For all D4RL environments, we follow the network architectures of CQL (Kumar et al., 2020) and IQL (Kostrikov et al., 2022), where a neural network of 2 encoding layers of size 256 is used, followed by an output layer. We use ELU activation function (Clevert et al., 2015) and SAC (Haarnoja et al., 2018) as the base RL algorithm. When approximating $\mathbb{E}_{\pi_\theta(a|s)}\left[e^{T_\psi(s,a)}\right]$, we use 50 Monte-Carlo samples. In addition, for unbiased gradient estimation with MCMC samples, we use a burn-in steps of 5. For all tasks, we average the mean returns over 10 evaluation trajectories and 5 random seeds. Detailed setups and hyper-parameters are in the appendix.

**Gym Locomotion Tasks.** We first evaluate MISA on the standard MuJoCo-style continuous control tasks, reported as gym-locomotion-v2 in Table 1. We observe that MISA improves the performance of baselines by a large margin. Specifically, MISA is less sensitive to the characteristics of data distributions. The medium datasets include trajectories collected by an SAC agent trained to reach 1/3 of the performance of an expert; the medium-replay datasets contain all data samples of the replay buffer during the training of the medium SAC agent, which covers the noisy exploration process of the medium agent. We can observe that prior methods are generally sensitive to the noisy sub-optimal data in medium and medium-replay environments, while MISA outperforms them by a large margin. In particular, MISA achieves near-expert performance on walker2d-medium-replay with only sub-optimal trajectories. This indicates that by regularizing the policy and Q-values within the mutual information of the dataset, we can fully exploit the data and perform safe and accurate policy improvement during RL. Moreover, on medium-expert environments, where the datasets are mixtures of medium agents and experts, MISA successfully captures the multi-modality of the datasets and allows further improvements of the policy over baselines.

**Adroit Tasks.** According to (Fujimoto & Gu, 2021), adroit tasks require strong policy regularization to overcome the extrapolation error, because the datasets are either generated by human (adroit-human-v0), which would show a narrow policy distribution, or a mixture of human demonstrations and a behavior cloning policy (adroit-cloned-v0). We observe that MISA provides a stronger regularization and significantly outperforms the baselines on adroit domains.

**Kitchen Tasks.** An episode of Kitchen environment consists of multiple sub-tasks that can be mixed in an arbitrary order. We observe that MISA outperforms baselines on both kitchen-complete-v0 and kitchen-mixed-v0, while achieving slightly worse performance on kitchen-partial-v0. Specifically, on kitchen-mixed, the result is consistent with our assumption that by better regularizing the policy, MISA guarantees a safer and in-distribution policy improvement step in offline RL.

**Antmaze Tasks.** On the challenging AntMaze domain with sparse delayed reward, we observe that MISA generally outperforms CQL and achieves the best performance on umaze environments. However, MISA performs worse than IQL on the challenging large environments. Multi-step value update is often necessary for learning a robust value estimation in these scenarios (Kostrikov et al., 2022) while MISA adopts a single-step SAC for the base RL algorithm.

| Dataset | $k$=5 | $k$=20 | BI=1 | no BA | BA | MISA-$f$ | MISA-DV | MISA-biased | MISA |
|---|---|---|---|---|---|---|---|---|---|
| halfcheetah-medium-v2 | 47.1 | 46.9 | 47.2 | **49.1** | 56.3 | 43.5 | 45.5 | 48.4 | 47.4 |
| hopper-medium-v2 | 62.2 | 65.3 | 61.8 | 64.4 | 1.2 | 60.5 | 61.6 | 65.7 | **67.1** |
| walker2d-medium-v2 | 83.3 | 83.9 | 81.8 | 83.8 | 7.5 | 73.2 | 82.8 | **84.2** | 84.1 |
| halfcheetah-medium-replay-v2 | 45.4 | 45.3 | 45.2 | 46.5 | **52.4** | 39.8 | 43.8 | 46.9 | 45.6 |
| hopper-medium-replay-v2 | 79.9 | 88.4 | 72.9 | **100.3** | 56.4 | 34.8 | 45.9 | 98.1 | 98.6 |
| walker2d-medium-replay-v2 | 83.7 | **86.9** | 82.8 | 86.1 | 51.1 | 34.9 | 81.4 | 80.6 | 86.2 |
| halfcheetah-medium-expert-v2 | 94.5 | 92.8 | 92.8 | 87.1 | 26.8 | 57.6 | 92.4 | 84.6 | **94.7** |
| hopper-medium-expert-v2 | 105.7 | 102.7 | 93.4 | 89.6 | 1.3 | 57.7 | **111.5** | 103.2 | 109.8 |
| walker2d-medium-expert-v2 | 109.2 | **109.4** | 109.3 | 108.1 | 1.4 | 102.7 | 108.8 | 109.2 | **109.4** |
| gym-locomotion-v2 (total) | 711 | 721.6 | 687.2 | 715 | 254.4 | 504.7 | 673.7 | 720.9 | **742.9** |

Table 2: Ablation studies on gym-locomotion-v2. $k$ denotes the number of Monte-Carlo samples for estimating $\mathbb{E}_{\pi_\theta(a|s)}\left[e^{T_\psi(s,a)}\right]$, BI represents the burnin-steps for MCMC simulation, and BA denotes the use of Barber-Agakov Bound. In addition, MISA-$x$ denotes different variants of MISA.

## E.2 Ablation Studies

To better understand MISA, we conduct extensive ablation studies on each component (Table 2).

*MISA requires careful Monte-Carlo approximation.* Firstly, we vary the number ($k$) of Monte-Carlo samples for approximating $\mathbb{E}_{\pi_\theta(a|s)}\left[e^{T_\psi(s,a)}\right]$ and reduce the burn-in steps of MCMC sampling process. Both operations would introduce additional Monte-Carlo approximation errors to MISA. Comparing $k=5$, $k=20$, and MISA ($k=50$), the performance increases monotonically; comparing MISA (BI=5) with BI=1, we can observe a sharp performance drop. We then conclude that MISA requires careful Monte-Carlo approximation for good performance.

*Accurately estimating the mutual information $I(S;A)$ is critical for offline RL.* In Table 2, BA stands for the Barber-Adakov Bound, while no BA stands for removing the Barber-Agakov term in Eqn. 7, which gives an inaccurate estimation (neither an upper bound nor a lower bound) to $I(S;A)$. We observe a clear performance drop when comparing them with MISA. In addition, as discussed in Sect. 2.2, considering the tightness of the various lower bound, we have BA $\leq$ MISA-$f$ $\leq$ MISA-DV $\leq$ MISA. Empirically, in Table 2, we observe that the overall performance of these four variants is consistent with the tightness of the bounds. This suggests that accurately estimating the $I(S;A)$ is crucial to offline RL and tighter bounds often give better performance.

*Unbiased gradient estimations improves performance of MISA.* Lastly, we study the importance of unbiased estimation discussed in Sect. 2.3. MISA-biased ignores the bias correction term in Eqn. 12. Although MISA-biased outperforms the baselines, it still performs worse than MISA. This suggests that by correcting the gradient estimation with additional MCMC samples, MISA achieves a better regularized policy learning in offline RL.

## E.3 Visualization of Embedding

In Fig. 1, we visualize the embeddings before the output layer of Q-value networks, given different mutual information bounds (BA and MISA). We select a subset from walker2d-medium-v2 dataset to study the division of low reward (blue) and high reward (red) $(s,a)$ pairs. We color each point by the reward $r(s,a)$. As discussed in Sect. 2.2, BA gives a lowest bound for mutual information estimation and MISA produces the tightest bound. In Fig. 1, we observe a consistent result. The embeddings of BA converge to a set of regular curves and fail to cluster the high $r(s,a)$, because Q-values have converged to indistinguishably high values ($3 \times 10^{12}$) for all $(s,a)$ pairs. In contrast, MISA successfully learns to cluster the $(s,a)$ pairs with a high reward into a cluster. From this perspective, we claim that regularizing the mutual information encourages learning a robust representation in offline RL scenarios.

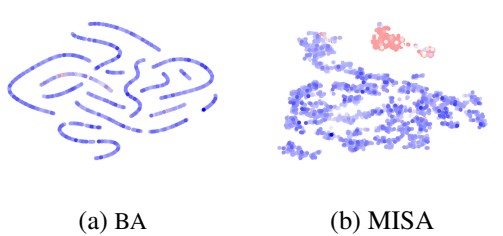

(a) BA          (b) MISA

Figure 1: tSNE of the Q-value network embeddings of walker2d-medium-v2 dataset, where red color denote high reward and blue color denote low reward.

### E.4 Limitations

Although MISA achieves great performance on several benchmarks as reported in Table 1, we have made the assumption that high mutual information comes from the stationary policy of a well-behaving agent. This will prevent MISA from being applied to tasks with extremely low-quality data, e.g., a random policy whose $I(S; A)$ is near zero. We validate this limitation by running the locomotion-random-

| Dataset | CQL | IQL | MISA |
|---|---|---|---|
| halfcheetah-random-v2 | **35.4** | 20.5 | 8.6 |
| hopper-random-v2 | **10.8** | 7.8 | 7.3 |
| walker2d-random-v2 | 7 | **8.9** | 2.2 |

Table 3: Results on locomotion-**random**-v2

v2 datasets of D4RL benchmark. The results are presented in Table 3. We observe that on datasets generated by random policies, MISA achieves worse performance than both CQL and IQL.

### E.5 Additional Experimental Details

For gym-locomotion-v2, kitchen-v0, and adroit-v0 environments, we average the results over 10 evaluation episodes and 5 random seeds. Following (Kostrikov et al., 2022), we evaluate the antmaze-v0 environments for 100 episodes instead. To stabilize the training of our agents in antmaze-v0 environments, we follow (Kumar et al., 2020) and normalize the reward by $r' = (r - 0.5) * 4$.

In addition, for a fair comparison with baseline methods, we use the same network structure as used in CQL (Kumar et al., 2020), where a network with embedding layers of sizes (256, 256, 256) is used for antmaze-v0 environments, and embedding layers of sizes (256, 256) is used for other tasks. ELU activation is used after each layer (Clevert et al., 2015). We use a learning rate of $1 \times 10^{-4}$ for both the policy network and Q-value network with a cosine learning rate scheduler. To sample from the non-parametric distribution $p_{\theta,\phi}(a \mid s) = \frac{\pi_\theta(a|s)e^{Q_\phi(s,a)}}{\mathbb{E}_{\pi_\theta(a|s)}\left[e^{Q_\phi(s,a)}\right]}$, we use Hamiltonian Monte Carlo algorithm. As MCMC sampling is slow, we trade-off its accuracy with efficiency by choosing moderately small iteration configurations. Specifically, we set the MCMC burn-in steps to 5, number of leapfrog steps to 2, and MCMC step size to 1.

For practical implementations, we follow the CQL-Lagrange (Kumar et al., 2020) implementation by constraining the Q-value update by a "budget" variable $\tau$ and rewrite Eqn. 9 as

$$\min_Q \max_{\gamma_1 \geq 0} \gamma_1 \left( \mathbb{E}_{s \sim \mathcal{D}} \left[ \log \mathbb{E}_{\pi_\theta(a|s)} \left[ e^{Q_\phi(s,a)} \right] \right] - \mathbb{E}_{s,a \sim \mathcal{D}} \left[ Q_\phi(s,a) \right] - \tau \right) - J_Q^{\mathcal{B}}(\phi). \qquad (25)$$

Eqn. 25 implies that if the expected value of Q-value difference is less than the threshold $\tau$, $\gamma_1$ will adjust to close to 0; if the Q-value difference is higher than the threshold $\tau$, $\gamma_1$ will be larger and penalize Q-values harder. We set $\tau = 10$ for antmaze-v0 environments and $\tau = 3$ for adroit-v0 and kitchen-v0 environments. For gym-locomotion-v2 tasks, we disable this function and direction optimize Eqn. 9, because these tasks have a relatively short horizon and dense reward, and further constraining the Q values is less necessary. Our code is implemented in JAX (Bradbury et al., 2018) with Flax (Heek et al., 2020) neural networks library.

