# OpenReview forum: "Mutual Information Regularized Offline Reinforcement Learning"
_NeurIPS.cc/2022/Workshop/Offline_RL — Offline RL Workshop NeurIPS 2022_

### Official Review · Reviewer_pSxh · 2022-10-07

**Rating:** 6
**Confidence:** 2

**Review:**

This paper proposes a novel MISA framework to approach offline RL from the perspective of Mutual Information between States and Actions in the dataset by directly constraining the policy improvement direction. Intuitively, mutual informa- tion measures the mutual dependence of actions and states, which reflects how a behavior agent reacts to certain environment states during data collection. To effectively utilize this information to facilitate policy learning, MISA constructs lower bounds of mutual information parameterized by the policy and Q-values.

This paper shows that optimizing this lower bound is equivalent to maximizing the likelihood of a one-step improved policy on the offline dataset. In this way, we constrain the policy improvement direction to lie in the data manifold.

I believe the result is good overall, however, why the standard deviation is not report in the table? Those uncertainty measure can further measure how good this MISA method is.

---

### Official Review · Reviewer_qwSj · 2022-10-19
**Interesting idea, good theory, and great results**

**Rating:** 9
**Confidence:** 3

**Review:**

This paper is well written and introduces a novel framework of incorporating a mutual information regularizer in the policy evaluation and policy improvement step. They show how this state-action mutual information regularizer framework is a generalization of other behavior-regularized approaches like TD3+BC and conservative-value approaches like CQL.

Experimentally they show that their method generally outperforms the relevant baselines on several of the D4RL tasks. Additionally, they include many insightful ablations on how different variants of their MISA algorithm performs.

Pros:
* Interesting idea that generalizes many previous SOTA offline RL approaches
* Well supported final practical algorithm
* Improves results on many D4RL tasks
* Good ablations

Cons:
N/A